# Knowledge Mapping for Fire Risk Assessment: A Scientometric Analysis Based on VOSviewer and CiteSpace

**Zhixin Tang, Tianwei Zhang \*, Lizhi Wu, Shaoyun Ren and Shaoguang Cai**

Hebei Key Laboratory of Emergency Rescue Technology, China People's Police University, Langfang 065000, China; zhixin0789@163.com (Z.T.); wulizhi119@souhu.com (L.W.); renshaoyun@cppu.edu.cn (S.R.); shaoguangcaicai@163.com (S.C.)
\* Correspondence: zhangtianwei@cppu.edu.cn

**Abstract:** Fire risk assessment is a crucial step in effective fire control, playing an important role in reducing fire losses. It has remained a significant topic in the field of fire safety. To explore the research hotspots and frontier trends in fire risk assessment and to understand its macroscopic development trajectory, a sample of 1596 papers from 1976 to 2023, extracted from the Web of Science (WoS) database, was utilized to create a knowledge map. The study employed bibliometric methods, visual analysis, and content analysis to uncover the research pulse and hotspots in the field, offering insights into its future development. The findings indicate that research in fire risk assessment has demonstrated continuous growth over the past 50 years. China and the United States are the dominant research forces in the field, while India and Australia show potential as new drivers for development. Expert groups have formed in this field, with intra-institutional cooperation being the primary focus, while inter-institutional collaboration remains limited. The research outcomes exhibit multidisciplinary crossovers, exerting a significant impact on various disciplinary domains. The research hotspots primarily revolve around investigating fire and explosion accidents, assessing the vulnerability of fire subjects, and identifying potential fire hazards. The application of artificial intelligence technology is identified as a pivotal tool for future development. However, to achieve substantial progress, it is important to enhance the importance accorded to fire risk assessment, foster multinational and cross-institutional cooperation, and prioritize research innovation.

**Keywords:** fire risk assessment; knowledge graph; VOSviewer; CiteSpace; bibliometrics



## 1. Introduction

Fire is a disaster caused by uncontrolled combustion in time or space [1], and it is a catastrophic problem faced by people worldwide. Its invisibility, danger, suddenness, and uncertainty pose a serious threat to public safety, social development, and ecological environment. Fire risk assessment refers to the process of analyzing the risk factors that influence the occurrence and development of fires through qualitative or quantitative methods, predicting the probability of fire occurrence and the consequences of the disaster, and subsequently recommending corresponding prevention and control measures. In recent years, large-scale fire accidents have occurred globally. For example, the Australian bushfires in 2019 killed nearly three billion animals [2]. On 4 August 2020, two explosions in the capital of Lebanon, Beirut, resulted in 73 deaths and 3000 injuries [3]. On 22 March 2021, a major fire broke out in a refugee camp in Bangladesh, resulting in 15 deaths, 560 injuries, 400 missing persons, and the destruction of nearly 10,000 shelters [4]. The significant hazards posed by fire accidents have raised awareness about the urgent need to study the patterns of fire occurrence and evolution, identify hidden risks, and improve fire prevention and control measures. Conducting a reasonable and accurate fire risk assessment is a prerequisite and foundation for solving this problem, and it has received sustained attention from an expanding global research community.

Currently, scholars in the field typically categorize the subjects of fire risk assessment, providing an overview and summary of research progress and cutting-edge developments in assessing fire risks in specific scenarios. For instance, Kong et al. systematically summarized the occurrence mechanisms, risk assessments, and prevention and control methods of coal spontaneous combustion, establishing an integrated system for coal spontaneous combustion prevention and control, while outlining future development directions for the field [5]. Ntzeremes et al. compared and sorted the risk assessment methods for road tunnel fires, highlighting the advantages and shortcomings of different methods, which effectively advanced research and progress in fire risk assessment methods for road tunnels [6]. In order to address the uncertainty associated with wildfire risk assessment and management, Thompson et al. compiled existing decision support systems and mapped the uncertainties of fire into appropriate decision-making tools, thereby enhancing the capabilities of human and ecological value in wildfire risk management [7]. Furthermore, Park et al. conducted a risk assessment of lithium-ion battery explosions, toxicological information on leaked chemicals, and the associated potential health risks, proposing advanced technical reference solutions to tackle the issue [8]; Pacifico et al. used two different types of fires as an example to collect and risk assess potentially toxic elements in fire ash, using robust principal component analysis and geospatial analysis to provide a feasible method for characterizing the composition of burning materials in fire events [9]; Dimitrios E. Alexakis summarized the results of major and micronutrient testing of ash from residential and wilderness areas burned by wildfires, discussed the potential hazards and regional distribution characteristics of wildfire ash, and assessed the health risks associated with wildfire ash to human health and terrestrial ecological receptors [10]. However, due to the wide range of disciplines and knowledge foundations involved in the field of fire risk assessment, organizing and analyzing it systematically is challenging. There are few studies that provide a comprehensive overview of the overall landscape of fire risk assessment from a global perspective and evaluate the hot frontiers and development trends in the field using bibliometrics and visualizations.

Bibliometrics, as a mature research method based on literature retrieval, has gained wide recognition in the academic community for analyzing key information about a specific field of literature, such as source journals, high-producing countries, organizations, etc. [11]. Currently, scholars have successfully applied bibliometric methods to various disciplines, including chemistry, biology, medicine, and disaster science, yielding notable achievements. For example, visual analysis has been conducted on the field of psycho-cardiology over the past 20 years [12], and CiteSpace and VOSViewer have been employed to analyze literature related to crowd evacuation [13]. Additionally, a quantitative analysis and systematic review of published articles in the field of electrochemiluminescence from 2000 to 2021 has been conducted [14], along with a discussion on research advances in the bioavailability assessment of polycyclic aromatic hydrocarbons (PAHs) and recent research priorities [15]. Bibliometrics enable a systematic analysis of the research landscape and the organization of existing theoretical frameworks, as well as providing strong support for constructing the knowledge foundations, exploring frontiers, and identifying development trends in a particular field. VOSViewer and CiteSpace are two kinds of commonly used visualization analysis software tools in bibliometrics. VOSViewer can provide the production of concise and clear network knowledge maps, mostly used for collaborative analysis and cluster analysis [16], while CiteSpace, developed by Chaomei Chen and his team, focuses on co-cited literature analysis, keyword analysis, and highly cited outbreak analysis [17]. Therefore, VOSviewer and CiteSpace have been applied in several research areas, but there is a lack of research based on these methods in the field of fire risk assessment.

Based on this, in order to understand the overall overview of the field of fire risk assessment, this paper uses two bibliometric software tools—VOSviewer and CiteSpace—to collect and organize the literature related to fire risk assessment from the WoS database. The aim is to systematically analyze the overall research landscape of this field from seven perspectives: chronological characteristics of the literature, high-influence authors, countries,

institutions, journals, subject terms, and cited literature. This analysis enables the visualization and examination of the literature's overview, hotspots, frontiers, and future trends in fire risk assessment. By doing so, it broadens the scientific perspective and provides valuable insights and references for further advancements in fire risk assessment research. The research framework is summarized in Figure 1.

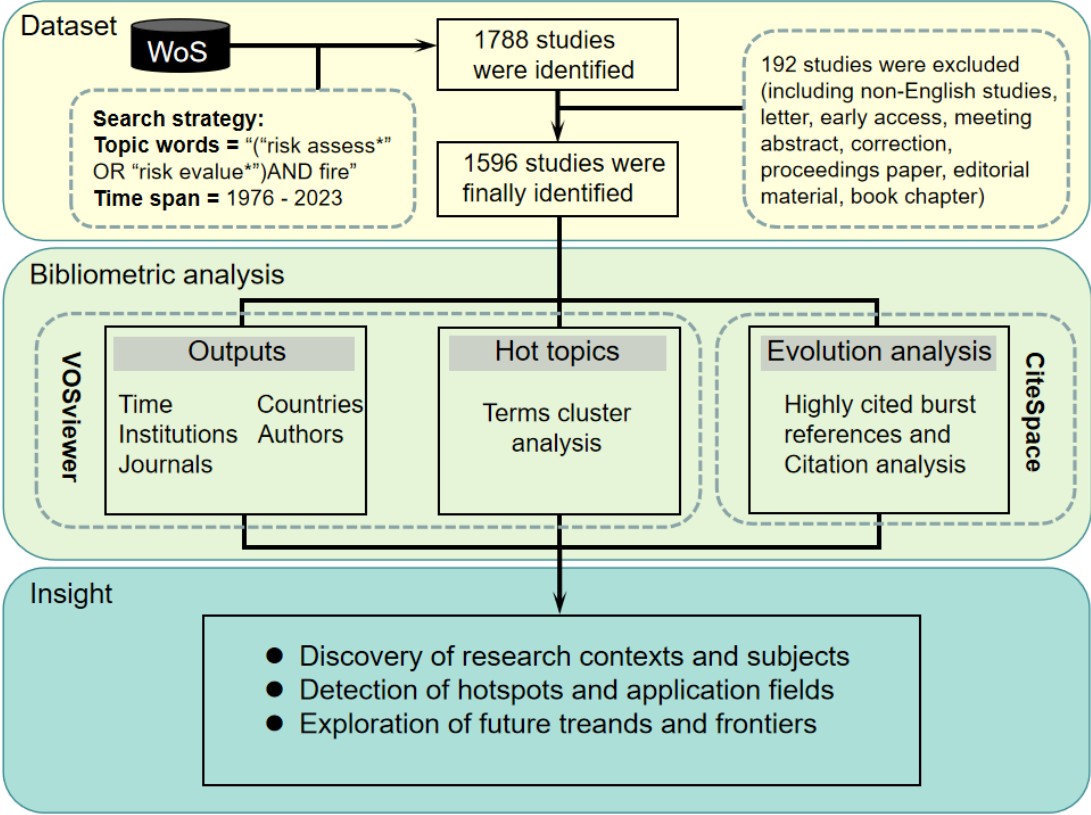

**Figure 1.** The summary of the flowchart and study design. (The wildcard (*) stood for plural and other forms of the words).

## 2. Data and Methods

### 2.1. Data Source

Web of Science is an internationally recognized database of scientific and technical literature, known for its high research standards. It covers a wide range of disciplines and includes influential journals and academic papers from various fields, making it a commonly used tool for literature search and journal evaluation. In this study, the WoS database was utilized for bibliometric analysis to ensure comprehensive, scientifically rigorous, and reliable data. The subject-matter search strategy was used to retrieve literature on "fire risk assessment" from papers indexed in the Science Citation Index Expanded (SCIE), using the search terms "("risk assess*" OR "risk evalue*") AND fire", with the wildcard (*) standing for plural and other forms of the words. The search was conducted on 12 April 2023, covering the period from 1976 to 2023 (with real-time updates for the 2023 search). The search yielded information on 1788 articles, and after manual screening, 192 irrelevant data points were excluded, resulting in 1596 data entries, including titles, authors, journals, abstracts, and references. These data were saved in plain text format for easy import and subsequent literature analysis.

When using the two software programs, VOSviewer1.6.19 and CiteSpace5.7r5, to generate knowledge graphs, the corresponding threshold parameters needed to be set for reasonable data screening and deployment. The data sorting results are generated based on empirical judgments concerning the volume of article data and the appropriate

configuration of threshold parameters. In cases where the results yield an excess or shortage of data, adjusting the respective threshold parameters allows for achieving an optimal amount of data results. In order to ensure that the obtained map is intuitive and clear, you can choose "Pathfinder" and "Pruning Visualization" for data standardization operations, and other functions such as "Burst Detection", "TimeZone" and other modules, to achieve the desired comprehensive display of the knowledge map.

### 2.2. Methods

Bibliometrics refers to the quantitative and statistical analysis of literature information and its temporal and spatial characteristics. It involves analyzing the quantity, distribution structure, and patterns of change in the existing literature in a research field, to uncover the discipline's evolutionary trends, research status, hotspots, and frontier trends [18]. Bibliometrics, which combines statistical and mathematical methods to rationally analyze the impact or value of research results, has become an important method in the research community. Content analysis is an objective, systematic, and quantitative descriptive research technique of clearly characterized communication content, applicable to both textual and non-textual forms of material. It is characterized by the continuous extraction and generalization of the required theory from a large amount of data. Combining bibliometric methods with content analysis in conducting a literature review of a research field can effectively address the limitations of relying solely on quantitative or qualitative analysis, thereby enhancing the reliability and accuracy of research findings. Data visualization employs computer image processing techniques to respresent data through graphical elements displayed on a screen, facilitating easy human interpretation and enabling interactive exploration. This approach offers diverse dimensions for flexible data observation and analysis, simplifying the extraction of crucial insights and enhancing the efficiency of data analysis. Knowledge mapping is an emerging approach in bibliometrics that utilizes visual representation to explore literature information to the fullest extent possible, thus providing a more comprehensive and intuitive guide for information services.

Therefore, based on the above three methods, this paper uses the VOSviewer1.6.19 and CiteSpace5.7r5 software tools to conduct statistical analyses on the selected literature regarding fire risk assessment. Scientific mapping analysis, collaborative network analysis, co-occurrence analysis, and cluster analysis are employed to uncover the overall landscape, knowledge structure, and development trends in the field of fire risk assessment, aiming to facilitate the sustainable development of future research in this area.

## 3. Results and Discussion

### 3.1. Research Status

3.1.1. Analysis of Publication Outputs

Studying the changes in the quantity and cumulative volume of literature related to fire risk assessment over time helps to grasp the research landscape and development trends in this scientific field. Figure 2 shows the temporal distribution of literature in the field of fire risk assessment, with the earliest publications dating back to 1976. Since then, there has been a consistent increase in the number of publications and the overall volume of literature, indicating a growing emphasis on research in fire risk assessment.

The trend in the number of publications in the field of fire risk assessment can be divided into three periods. From 1976 to 1993, there is low research focus and minimal academic output in fire risk assessment, with fewer than three publications per year and even some years without any publications, indicating the nascent stage of this field. The period from 1994 to 2005 sees a cumulative total of 156 publications, showing a significant increase compared to the previous decade. This indicated that the field started to receive attention from the research community and entered a phase of stable exploration. However, the distribution of publications is uneven, with the highest number of publications occurring in 2002, which is closely related to the forest fires in Australia at the end of 2001 [19]. These fires captured global attention by setting records for the highest number of fire incidents

and the largest burnt areas in the past 50 years. From 2006 to 2023, there is a continuous emergence of relevant literature, with a substantial increase in the number of publications. This period can be characterized as a period of rapid development, especially after China hosted the 8th International Conference on Fire Science in 2005, which led to an increase in the number of articles published by Chinese scholars in international journals, aligning the level of fire research in China with international standards. A total of 1426 articles are published during this period, accounting for 89.3% of the total number of articles (1596), with a peak of 185 articles in 2022. This represents the in-depth and extensive research conducted worldwide in the field of fire risk assessment in recent years, driving the field towards gradual maturity and the possibility of reaching new heights in the coming years. Overall, fire risk assessment has consistently been a hotspot in fire science research, accumulating a substantial body of research.

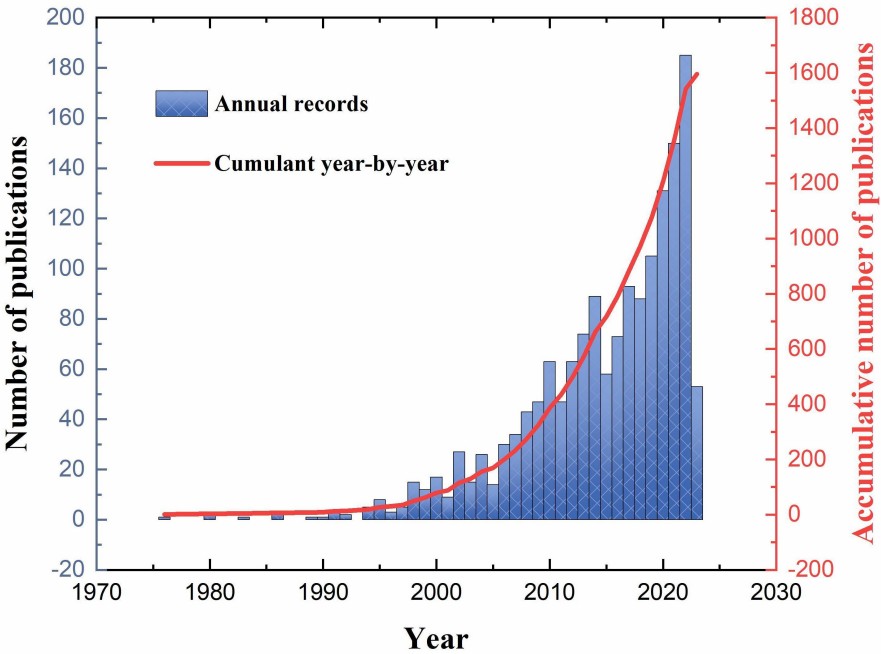

**Figure 2.** Number of publications in each year from 1976 to 2023.

3.1.2. Analysis of Countries

To investigate the contributions of different countries and regions to the field of fire risk assessment, Figure 3 displays a collaborative network of countries with more than 15 publications, based on a time distribution. In terms of the number of publications by country, China and the USA hold leading positions in the field of fire risk assessment. China has published 528 articles, accounting for 33% of all publications, followed by the USA with 231 articles. Together, these two countries contribute to half of the total number of articles published. Regarding the average number of citations, the top three countries are the USA, China, and Spain. The partnership graph illustrates the frequency of collaborations between different countries. The most frequent collaborations are observed between the USA and China, with 20 collaborations each, followed by the UK with 19 collaborations.

In the field of fire risk assessment, China and the United States are the primary contributors, ranking high in terms of publication count, citations, and collaborations, and holding significant scientific influence in the field. Regarding collaborations, China and the USA not only have collaborations with multiple countries but also exhibit the strongest collaborative relationship with each other. Following them are European countries, led by Spain, Italy, and Germany, as well as Oceanian countries, led by Australia, and North American countries like Canada, demonstrating cross-regional collaborations. Most of the other leading countries in research are developed nations. This can be attributed to the fact that developed countries often possess advanced instruments, electronics, and

other facilities, which facilitate their research on fire characteristics, thermodynamics, and fire-resistant materials. Additionally, their robust economic foundations allow for a greater emphasis on disaster prevention and system security, resulting in more in-depth and extensive studies on fire risk assessment.

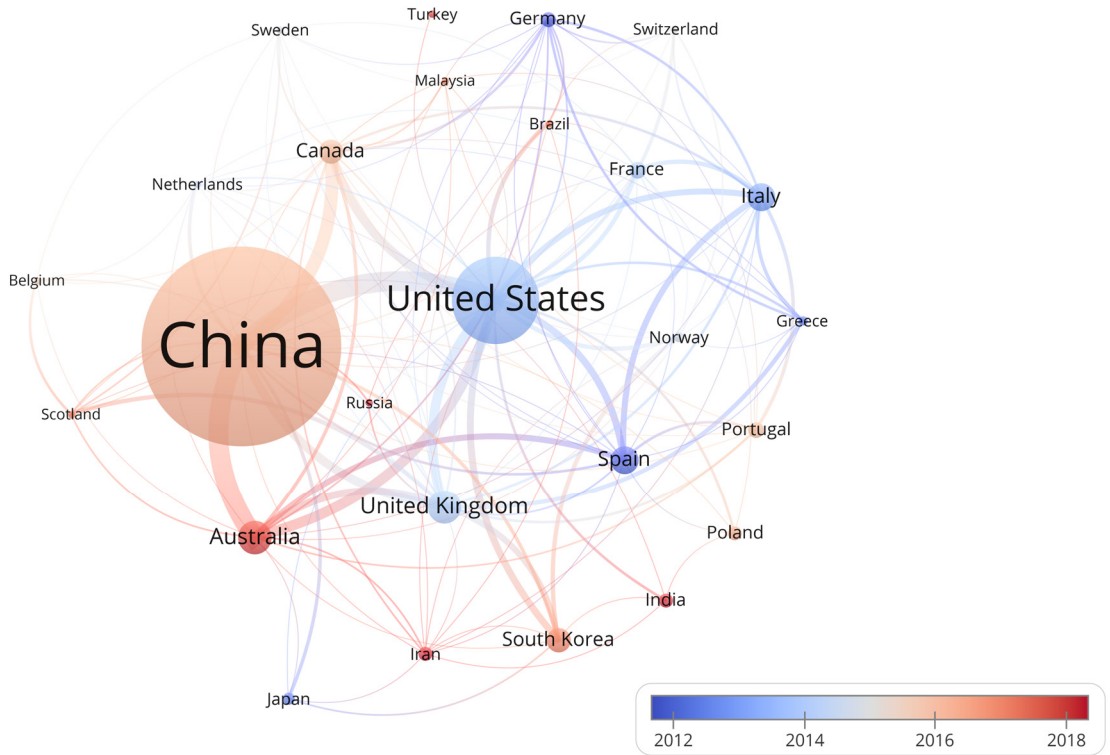
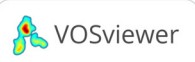

**Figure 3.** Countries/territories collaboration network of the research on fire risk assessment.

Compared to the USA and other European countries, China, along with Australia, Iran, and India, has conducted research relatively recently. China, with its high population density and susceptibility to fires, places significant importance on the field of fire risk assessment and has published a substantial number of relevant articles. While India also has a large population, its limited economic resources have hindered research in the area of fire risk assessment. However, as social and economic development progresses and fire-related issues become more pronounced, it is likely that India will increasingly focus on this field. Australia, highly susceptible to fires influenced by climate change, experienced its most severe fire season in history in 2019. This devastating event resulted in the loss of at least 15 lives, the destruction of 100 homes, and tens of thousands of square kilometers of land engulfed by fire. Consequently, Australia will continue to prioritize research in the field of fire risk assessment in the coming years.

### 3.1.3. Analysis of Institutions

The study sample consists of 1768 institutions and 96 countries. Figure 4 illustrates the institutions that have published more than 10 papers. In terms of publication count, the University of Science and Technology of China (USTC) has the highest number of papers, with a total of 38, accounting for 2.4% of the total. The US Forest Service ranks second with 33 publications, while the University of Mining and Technology of China (UMC) ranks third with 29 publications. In terms of average citation count, the US Forest Service has an average of 1444 citations, followed by USTC, and Memorial University of Newfoundland ranks third. In terms of collaborative relationships, the USTC collaborates with six other institutions. Additionally, the Chinese Academy of Sciences, Tsinghua University, City

University of Hong Kong, and the China University of Mining and Technology have extensive collaborative relationships.

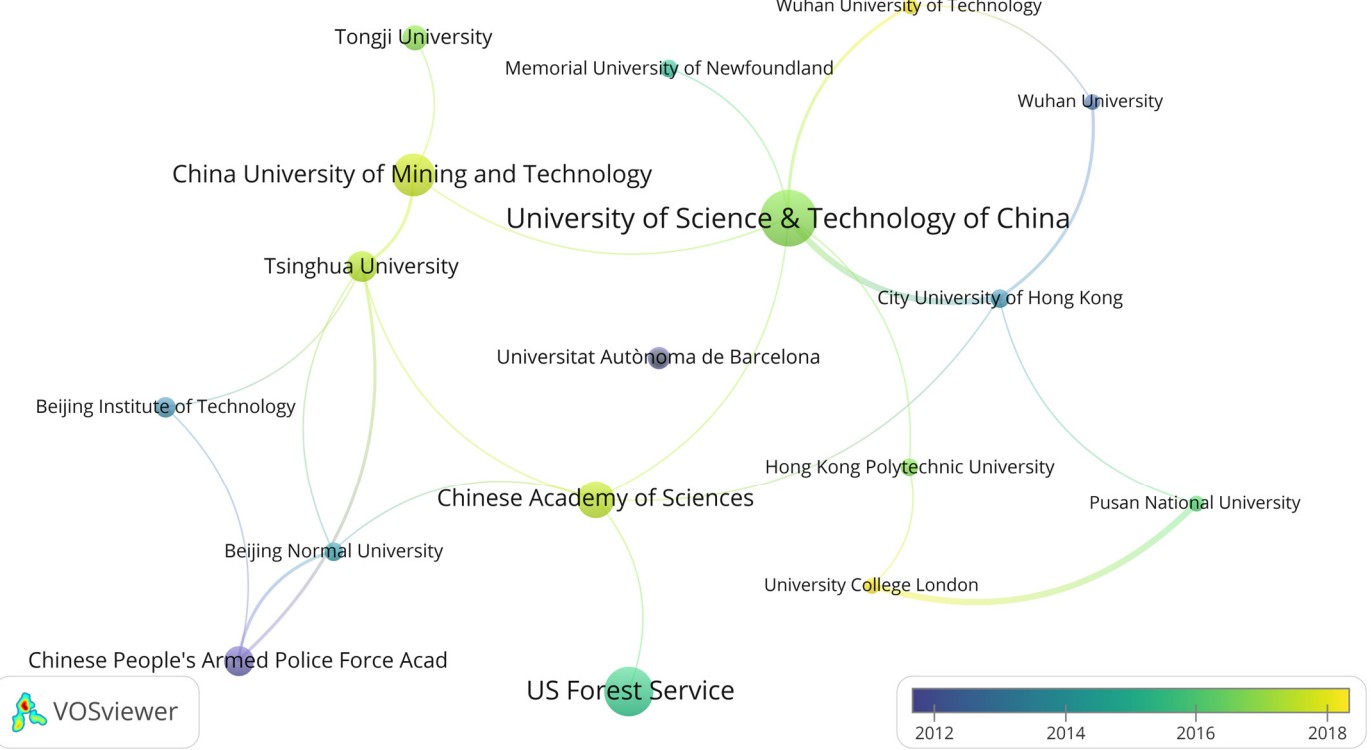

**Figure 4.** Institution collaboration network of research on fire risk assessment.

Among the top 10 institutions in terms of publication count, eight are from China, one is from the US, and one is from Spain. This indicates that China possesses a significant number of research institutions and publications, establishing it as the primary driving force in the field of fire risk assessment. However, in terms of average citation count, the US holds a leading position in advancing this field. Despite having a higher publication count and citation rate, China is continuously improving to narrow the gap. The US Forest Service has accumulated a total of 1444 citations for articles published in this field, demonstrating its substantial influence in fire risk assessment. Three articles from the US Forest Service have citations exceeding 200, with two of them authored by the Finney team, focusing on quantitative and simulation methods for wildfire risk assessment [20,21]. Another article by Donato et al. explores and analyzes the impact of post-wildfire logging behavior on regeneration fire risk [22]. These three articles hold significant value in the field of wildfire risk assessment.

It is worth mentioning that Memorial University of Newfoundland has the second-highest average number of citations, trailing only behind the University of Science and Technology of China. Despite its lower publication count, it has a high number of citations, indicating its potential to drive advancements in the field. The University of Science and Technology of China, being the institution with the most frequent collaborative contacts, tends to focus on research related to lithium-ion battery fires, flame retardant materials, deflagration, and fire propagation characteristics, using experimental or simulation methods. The Chinese People's Armed Police Force Academy, Beijing Institute of Technology, Beijing Normal University, University Autónoma de Barcelona, and Wuhan University of Technology were among the first institutions to initiate research on fire risk assessment. On the other hand, China University of Mining and Technology, Tsinghua University, and the Chinese Academy of Sciences represent emerging research forces in this field.

Regarding overall collaboration, there is close cooperation among institutions within China. However, international collaboration among institutions is lacking. It is essential for countries to engage in mutual communication and strengthen collaborative links to promote the further development of fire risk assessment.

### 3.1.4. Analysis of Authors

The distribution of high-impact authors yields valuable insights, uncovering the relationships among the most productive and influential authors in the field of fire risk assessment on WoS. It sheds light on the research focus of leading researchers. Figure 5 illustrates the co-occurrence distribution (a) and time-based network of collaborative relationships (b) among researchers in the field of fire risk assessment. To ensure clarity, only author teams with more than five collaborative relationships are included in the plot. The node size represents the average number of citations for articles authored by each author, the thickness of connecting lines indicates the strength of the collaborations, and the shades of color represent the temporal evolution of research orientations.

In terms of publication count, Ana Cortés and Faisal Khan emerge as the most prolific authors in the field of fire risk assessment, with each having published 12 papers. They are followed by Emilio Chuvieco and Tomas Margalef, with 10 publications each. When considering the average number of citations, Emilio Chuvieco, Marta Yebra and Juan De la Riva rank among the top three, with 1312, 834, and 556 citations, respectively. Regarding the number of co-authored papers, according to calculations using VOSviewer1.6.19 software, the top five authors in terms of collaboration frequency are Ana Cortés, Tomas Margalef, Mónica Denham Germán Bianchini and Emilio Luque.

The distribution of high-impact authors allows for a better understanding of the leading researchers and their contributions in the field of fire risk assessment, facilitating collaboration and knowledge exchange for further advancements.

The four most prolific authors in the field of fire risk assessment have different research interests. Ana Cortés focuses on research about forest fire data modeling and fire spread prediction algorithms [23,24]. Faisal Khan primarily investigates risk analyses of safety systems in marine operations, using operations research, statistics, and computer science [25–27]. Emilio Chuvieco specializes in large-scale fire assessments, utilizing geographic remote sensing, imaging techniques, and machine learning algorithms [28,29]. Meanwhile, Marta Yebra concentrates on exploring the relationship between vegetation, the environment, and the impact of forest fires [30]. The authorship map reveals four prominent authorship teams led by Tomas Margalef, Ana Cortés, and Emilio Chuvieco. These teams have a substantial number of publications in the field, primarily focusing on forest fire prediction and management [31,32]. It should be noted that, although these authors have achieved high publication and citation numbers, their articles are relatively old and may differ from current research trends. On the other hand, Asif Usmani and their team represent an emerging research group with strong collaborative relationships. They primarily study the risk assessment, prevention, and management of fires in various scenarios [33]. While there are some specialized research groups and cross-institutional collaborations in the field of fire risk assessment, the number of research teams remains limited, and their research focuses differ from one another.

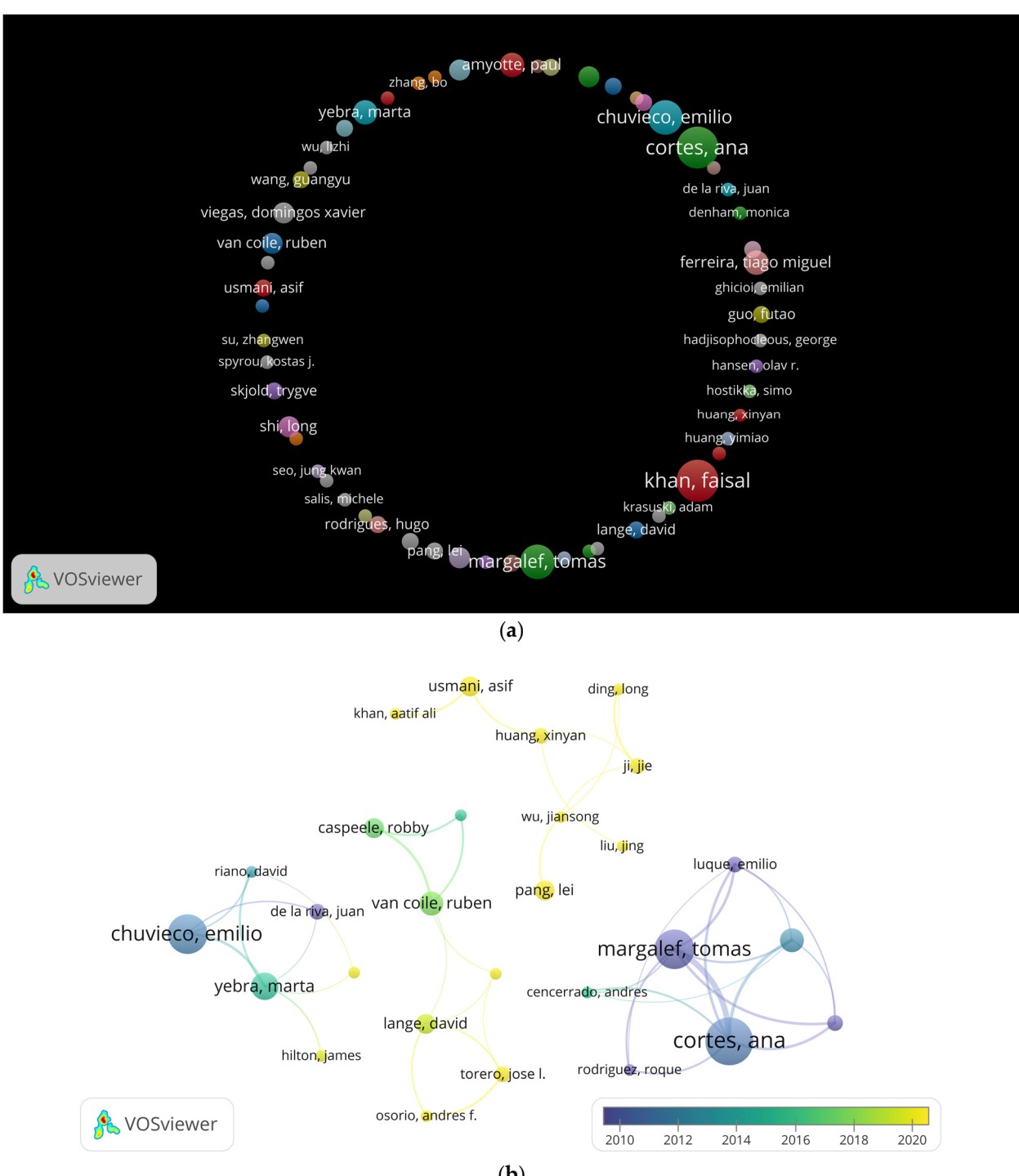

**Figure 5.** The non-interconnected cited author network (**a**) and the collaboration network of different authors (**b**).

### 3.1.5. Analysis of Journals

There are a total of 656 journals that publish research on fire risk assessment. Table 1 presents the top 10 journals ranked by the number of publications, including the journal names, publication count, 2021 impact factor (IF), 5-year impact factor, journal category, ranking, and division. These indicators are used to analyze the professional standing and level of influence of journals in the field, providing researchers with a reference for literature sources.

**Table 1.** The top 10 prolific journals in fire risk assessment, 1976 to 2023.

| Rank | Source Journal | Publications | IF-2021 | 5-Year IF | Journal Category | Quartile Rank | |
|------|----------------|--------------|---------|-----------|------------------|----|----|
| 1 | *Journal Of Loss Prevention In The Process Industries* | 62 | 3.916 | 3.857 | ENGINEERING, CHEMICAL | 62/143 | Q2 |
| 2 | *Fire Safety Journal* | 47 | 3.78 | 4.131 | ENGINEERING, CIVIL; MATERIALS SCIENCE, MULTIDISCIPLINARY | 49/138 174/345 | Q2 Q3 |
| 3 | *Fire Technology* | 35 | 3.605 | 3.276 | ENGINEERING, MULTIDISCIPLINARY; MATERIALS SCIENCE, MULTIDISCIPLINARY | 30/92 182/345 | Q2 Q3 |
| 4 | *International Journal Of Wildland Fire* | 33 | 3.398 | 3.783 | FORESTRY | 12/69 | Q1 |
| 5 | *Process Safety And Environmental Protection* | 27 | 7.926 | 7.717 | ENGINEERING, CHEMICAL; ENGINEERING, ENVIRONMENTAL | 21/143 13/54 | Q1 Q1 |
| 6 | *Fire-Switzerland* | 26 | 2.726 | 3.456 | ECOLOGY; FORESTRY | 93/173 22/69 | Q3 Q2 |
| 7 | *Remote Sensing* | 26 | 5.349 | 5.786 | ENVIRONMENTAL SCIENCES; GEOSCIENCES, MULTIDISCIPLINARY; IMAGING SCIENCE & PHOTOGRAPHIC TECHNOLOGY REMOTE SENSING | 83/279 30/202 6/28 11/34 | Q2 Q1 Q1 Q2 |
| 8 | *Fire And Materials* | 21 | 1.979 | 2.226 | MATERIALS SCIENCE, MULTIDISCIPLINARY | 266/345 | Q4 |
| 9 | *Process Safety Progress* | 16 | 1.294 | 1.249 | ENGINEERING, CHEMICAL | 110/143 | Q4 |
| 10 | *Forest Ecology And Management* | 15 | 4.384 | 4.584 | FORESTRY | 6/69 | Q1 |

The types of journals can be categorized into three main categories. Engineering safety journals, such as the Journal of Loss Prevention in the Process Industry, publish the highest number of articles. These journals primarily focus on safety issues related to fire, explosions, and toxic releases in industrial operations, emphasizing the safety of high-risk sites. Other examples include Process Safety and Environmental Protection and Process Safety Progress. Fire safety journals, like the Fire Safety Journal, rank second and specialize in fire risk assessment. They cover various aspects of fire safety engineering and include diverse articles related to fire protection. Additional journals in this category include Fire Technology, International Journal of Wildland Fire, Fire-Switzerland, and Fire and Materials. Forest fire prevention and control journals, such as Forest Ecology and Management, concentrate on articles related to forest fire prevention and control and post-disaster ecological damage. Another example is Remote Sensing. The distribution of journal disciplines and impact factors indicates that research in the field of fire risk assessment exhibits interdisciplinary characteristics. Furthermore, the articles generally maintain high quality and possess influence within different academic domains.

*3.2. Research Hotspots and Frontiers*

3.2.1. Analysis of Terms

Topic word clustering analysis can further the exploration of the intrinsic relationships between topic words, providing insights into the overall research landscape in the field of fire risk assessment. Using the bibliographic function of the VOSviewer1.6.19 software, subject terms are extracted from the titles and abstracts of 1596 documents, and a visual knowledge map of subject term clustering is created to demonstrate the temporal distribution of these terms (see Figure 6). The subject terms included in the knowledge map needed to appear in at least 35 documents, and the top 60% most relevant subject terms were selected by default, resulting in a total of 187 subject terms comprising the fire risk

assessment knowledge map. Overall, significant progress has been made in research within the field of fire risk assessment, with the research network taking shape and being divided into four clusters, represented by different colors: Cluster #1 "Typical fire site risks" (green), Cluster #2 "Fire risk assessment methodology" (yellow), Cluster #3 "Forest and regional fire assessment and prediction" (red), and Cluster #4 "Fire experiments and testing" (blue). These four clusters demonstrate a nearly symmetrical, compact, and balanced distribution, reflecting the equilibrium in the research priorities of the field of fire risk assessment. Table 2 provides the top 10 subject terms in each cluster.

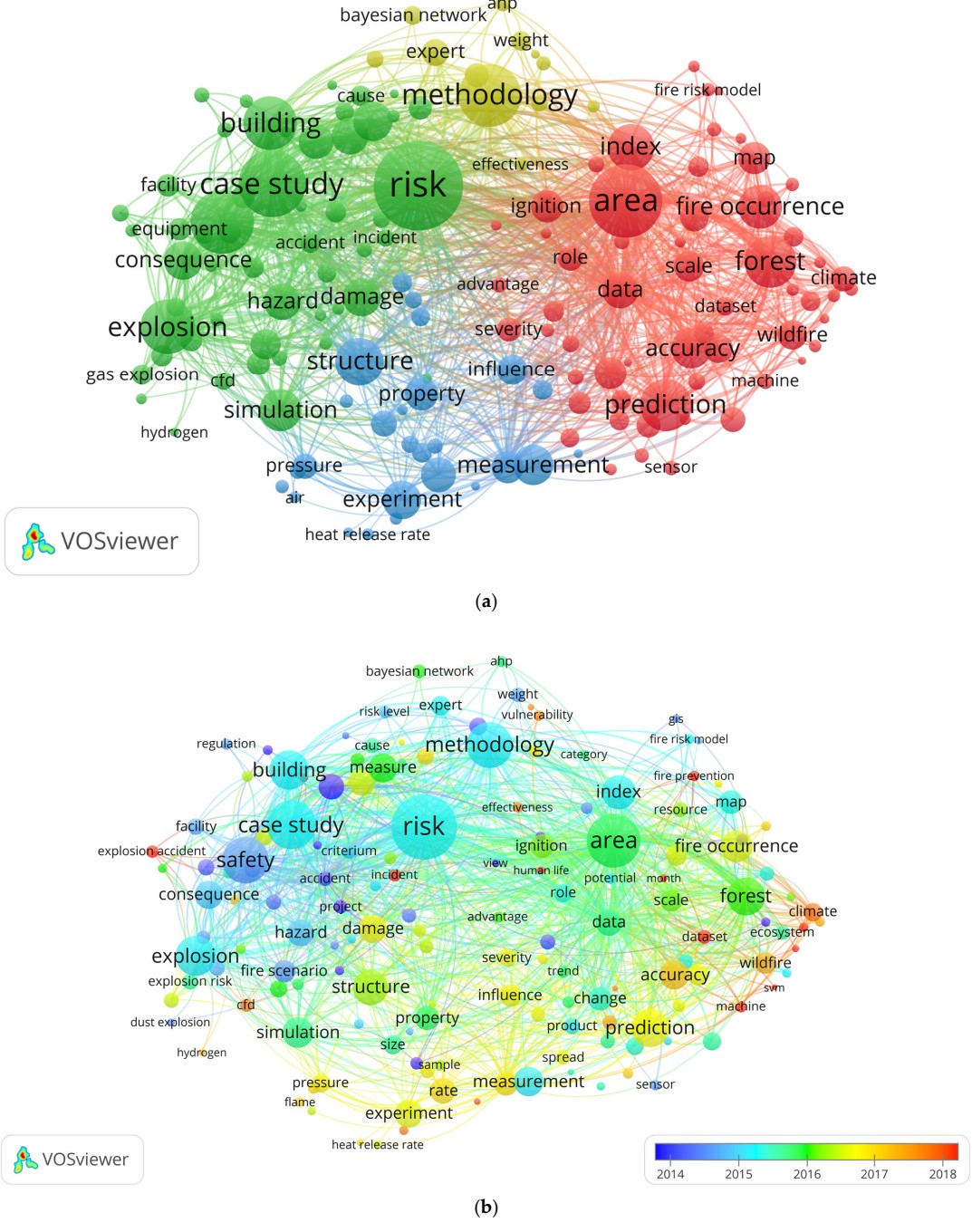

(**a**)

(**b**)

**Figure 6.** Terms cluster of the research on fire risk assessment: (**a**) distinguished clusters; (**b**) terms cluster by average year distribution.

**Table 2.** High frequency subject headings for fire risk assessment.

| Cluster#1 | | Cluster#2 | | Cluster#3 | | Cluster#4 | |
|---|---|---|---|---|---|---|---|
| Topics | Weights | Topics | Weights | Topics | Weights | Topics | Weights |
| risk | 470 | methodology | 301 | area | 368 | structure | 217 |
| case study | 330 | expert | 106 | forest | 239 | measurement | 171 |
| safety | 305 | decision | 90 | prediction | 238 | experiment | 161 |
| explosion | 251 | weight | 70 | index | 205 | rate | 143 |
| building | 248 | Bayesian network | 68 | fire occurrence | 192 | property | 137 |
| simulation | 182 | city | 61 | accuracy | 174 | temperature | 132 |
| measure | 167 | risk level | 61 | data | 150 | influence | 120 |
| damage | 165 | ahp | 53 | change | 133 | size | 102 |
| consequence | 164 | vulnerability | 48 | map | 130 | pressure | 95 |
| hazard | 155 | effectiveness | 47 | ignition | 130 | difference | 72 |

In Cluster #1, "Typical fire site risks", the theme word "risk" has the highest frequency, connecting 150 nodes and appearing 470 times. It is followed by "building" and "explosion", indicating that the research focus in fire risk assessment lies in building fires and fires caused by explosions. For instance, Wang et al. assessed the fire risk in large commercial and high-rise buildings using the fuzzy analytic hierarchy process and coupling revision, with the aim of evaluating the performance of their fire safety systems [34]. Pang et al. classified the severity and sensitivity of dust explosions based on different dust parameters, enabling the assessment of the explosion risk of polyethylene dust [35]. The terms "accident", "safety", and "case study" mainly appear in articles analyzing typical cases of specific fire types or determining risks. For example, Pula et al. used a grid-based approach to model and analyze the radiation and overpressure explosion risks caused by offshore conditions, providing a theoretical basis for implementing corresponding protective measures, reducing fire hazards, and ensuring personnel safety [36]. Cluster #1 also includes subject terms such as "simulation", "equipment", "damage", "consequence", and "hazards", reflecting the various risk factors in industrial settings that may lead to fires and the common technical approaches employed in risk assessment. For example, Yan et al. developed an analytical-based model for assessing fire risks in subway systems, incorporating passenger distribution simulation to enhance objectivity in the assessment and conducting in-depth research on fire risk assessment and control measures in subways [37].

Cluster #2, "Fire risk assessment methodology", revolves around the theme word "methodology", indicating that fire risk assessment methods are the primary focus of research in this branch. Other subject terms include "experts", "weights", "Bayesian network", and "AHP", which are common methods used in assessments. For instance, Khan et al. employed an analytic hierarchy process to evaluate the social and economic impact factors, structural vulnerability, and likelihood of bridge fires, constructing a bridge fire risk assessment model to provide a theoretical foundation for highway engineers designing fire protection structures for bridges [38]. The Bayesian network approach is a quantitative assessment method that involves the use of probability distributions and graph theory to create a directed acyclic graph, often used in conjunction with incident fault tree analysis in the field of fire risk assessment to illustrate the intricate network relationships between influencing factors. Pei et al. developed a fire risk assessment model for high-rise buildings that combined fault trees analysis and a Bayesian network, utilizing the model to calculate the probability and extent of loss at different stages of fire development, thereby demonstrating the practicality and reliability of the model [39].

In Cluster #3, "Forest and regional fire assessment and prediction", "area", "forest," "prediction", "index", and "fire occurrence" are the top five thematic terms in this cluster, indicating that it primarily focuses on regional and forest fires, with prediction being the main assessment method employed. The commonly used technique in this field is geographic information remote sensing technology. Other frequently mentioned thematic

terms include "ignition", "data", "map", and "accuracy". This suggests that articles in this category often use data for predicting and assessing the risk of regional wildfires or forest fires, visualizing fire points and risk levels on maps, and employing machine learning algorithms to ensure accurate predictions. For instance, Nami et al. combined a geographic information system (GIS) with evidential belief function (EBF) models to predict the spatial distribution of wildfires in a part of the Hyrcanian ecoregion of northern Iran [40]. The model was trained and validated using historical survey data and the MODIS hot spot product, confirming its reliability and effectiveness. Tien et al. used GIS and kernel logistic regression algorithms to establish a tropical forest fire susceptibility model [41]. Historical forest fire data were collected for training and prediction, resulting in an accuracy of 92.2%, surpassing the baseline model (SVM). This model proved valuable for forest fire management by local authorities.

The most commonly observed thematic terms in Cluster #4, "Fire experiments and testing", are "structure", "measurement", and "experiment", indicating that the primary focus of this branch is related to combustion experiments. Other frequently occurring thematic terms include "temperature", "influence", "pressure", "size", and "heat release rate". These are common physical quantities that need to be measured during combustion experiments. By measuring these physical quantities, it becomes possible to gain intuitive insights into the combustion characteristics and toxicological properties of different materials and their products. This process also allows for the development of empirical models, which provide a theoretical foundation for the quantitative assessment of fire risks. For example, Shi et al. conducted research using a cone calorimeter to investigate the production of carbon monoxide from 50 wood samples under spontaneous combustion [42]. The study explored the impact of various factors on the rate of carbon monoxide release and formulated empirical equations under different conditions. This research contributed to the establishment of a knowledge base for fire risk assessment modeling. Additionally, the measurement of these physical quantities enables a deeper understanding of the development and evolution of fire combustion and smoke. This knowledge plays a crucial role in guiding fire defense, design, and fire suppression strategies. For instance, Yuan et al. focused on four full-size wooden structures and employed equipment monitoring techniques and statistical methods [43]. The study explored the influence of factors such as relative slope, thermal radiation, and wind direction on fire spread trends. By investigating the fire spread characteristics and mechanisms of internal fires in full-size wooden houses, the research provided valuable insights for fire prevention, design, and firefighting efforts.

Figure 6b shows the distribution of the time-based clustering of the subject terms, with colors ranging from blue to red representing the evolution of the occurrence years of the terms from earlier to more recent. Overall, cluster #1, "Typical fire site risk", and cluster #2, "Fire risk assessment methodology", demonstrate a higher level of maturity, encompassing both earlier research content and newer research directions. On the other hand, cluster #3, "Forest and regional fire assessment and prediction", and cluster #4, "Fire experiments and testing", are at the forefront of fire risk assessment research, comprising numerous cutting-edge research topics.

Cluster #1's recent research has focused primarily on topics such as "explosive accident", "CFD", "incident", and "reliability". This branch explores the intersection of simulation and fire risk assessment [44], as well as conducting in-depth studies on fire and explosion accidents [45].

Cluster #2's recent research directions primarily revolve around themes such as "vulnerability", "decision making", "effectiveness", "city", and "prevention." When applying various fire risk assessment methods, the emphasis lies on understanding the vulnerability of fire subjects [46] and the effectiveness of the methodologies [47] to prevent fire accidents.

Cluster #3's emerging research directions focus on topics including "dataset", "climate", "fire prevention and control", "human life", "SVM", and "machine". Due to the difficulties in controlling fire sources, the complex factors within fire zones, and the challenges in fire suppression, historical data are often utilized for prediction and assessment

in the study of forest fires and regional fires. Artificial intelligence techniques, particularly machine learning and deep learning algorithms, are renowned for their high accuracy. Researchers have applied these algorithms to various fields of study, including fire risk assessment, assisting in areas such as fire case statistics [48], fire risk prediction [49], and fire monitoring and detection [50]. Support Vector Machines (SVMs) are commonly employed in fire prediction among machine learning algorithms [51], demonstrating significant advantages in fire detection. Furthermore, numerous scholars engaged in fire risk assessment have identified human activities and climate as influential factors that frequently contribute to forest fires or regional fires [52].

Cluster #4's recent research directions primarily encompass topics such as "high temperature", "severity", "endanger", and "influence". This highlights the attention given by the academic community to the impact and hazards of fire [53]. Exploring these aspects constitutes an essential component of fire theory, requiring researchers to delve deeper into the subject matter.

3.2.2. Analysis of Highly Cited Burst References

Highly cited burst literature refers to scholarly articles that experience a sudden increase in citations within a short period, capturing the attention and interest of the academic community at that time [54]. Analyzing highly cited burst literature aids in understanding the current hotspots and frontiers of the research field, as well as predicting future research directions and trends. The burstness module of CiteSpace is employed to analyze the highly cited bursts of literature, and the findings are presented in Table 3. The table displays the top 10 highly cited burst literature, along with their burst intensity, start and end time, and other relevant information.

Due to the short duration of the nascent and stable exploration periods, as well as the low number of publications, the highly cited outbreak literature in the field of fire risk assessment from 1976 to 2023 was concentrated in the rapid development period. The article with the highest outbreak value is Westerling et al. (2006), with an outbreak value of 8.73. Westerling et al. compiled a database of large wildfires in western US forests from 1970 to 2006 and compared it with hydrological, climatic, and surface data from the same period [55]. They found that the greatest increase in wildfire frequency occurred in mid-elevation Northern Rocky forests, with a close association with elevated spring and summer weather. Chuvieco et al. [56] and Martinez et al. [57] rank second and third in terms of outbreak value. Their research contents have similarities, as they both employed remote sensing and geographic information systems (GIS) to assess the spatial and temporal dimensions of fire risk in a specific region, providing reference data for fire prevention planning in that area. These three prominent outbreak articles have already highlighted the key issues in fire risk assessment, with data and methods being crucial entry points in this field, although the research is still in its early exploratory stages.

Regarding the timing of outbreaks, recent highly cited outbreak articles include Giglio et al. [58], Giglio et al. [59] by the Giglio team, and Guo et al. [60]. In these articles, Giglio's team extensively discussed improvements to fire detection algorithms. The former utilized MODIS fire point data to address false alarms caused by small forest clearings and missing data from large fires obscured by smoke. The latter focused on enhancing the detection accuracy of small fires, reducing uncertainty in burn times, and significantly minimizing the extent of unmapped areas. Guo et al. employed a logistic regression model and random forest algorithms to explore the causal factors of fire occurrence within Fujian Province, China. Their study assessed fire risk in the region, revealing climatic factors as the primary drivers. They also created a distribution map of high-risk fire areas, providing valuable guidance for fire prevention and control in the region. Both methods leverage the rapid advancements in computer technology, offering new research directions in the field of fire risk assessment. For scholars who are new to this field, reading highly cited outbreak literature from different periods enables them to quickly comprehend the technical advancements and identify important research periods and hot topics.

**Table 3.** Top 10 references with the strongest citation bursts.

| Ref. | Burst | Duration | Range (1976–2023) |
|---|---|---|---|
| Chuvieco, E., 2004 | 5.12 | 2008–2012 | |
| Chuvieco, E., 2010 | 6.97 | 2010–2018 | |
| Westerling, A.L., 2006 | 8.73 | 2012–2014 | |
| Martinez, J., 2009 | 6.72 | 2012–2017 | |
| Krawchuk, M.A., 2009 | 5.23 | 2013–2017 | |
| Padilla, M., 2011 | 5.32 | 2014–2018 | |
| Oliveira, S., 2012 | 6.01 | 2016–2020 | |
| Giglio, L., 2016 | 4.98 | 2020–2023 | |
| Giglio, L., 2018 | 5.26 | 2021–2023 | |
| Guo, F.T., 2016 | 5.19 | 2021–2023 | |

## 4. Conclusions

By analyzing 1596 papers in the field of fire risk assessment from the WoS database between 1976 and 2023, this study provides a systematic review of the research progress made in the past 50 years. The methodology employed includes bibliometric methods, visual analysis, and content analysis, enabling effective information extraction and knowledge mapping. The objective is to examine the evolution of research in fire risk assessment and suggest future research directions. The main findings are as follows:

(1) In terms of temporal development, the publication volume can be divided into three phases: the nascent period (1976–1993), the stable exploration period (1994–2005), and the rapid development period (2006–2023). Overall, there has been a continuous upward trend, reflecting the significance and ongoing attention of the academic community towards the field of fire risk assessment. In regard to spatial distribution, China and the United States hold dominant positions in driving the development of fire risk assessment, and they have established collaborative relationships with other countries. Additionally, India and Australia are emerging forces that have the potential to contribute to the advancement of research in this field.

(2) The University of Science and Technology of China, the US Forest Service, and the China University of Mining and Service are the key players among research institutions in the field of fire risk assessment. There are several expert research groups in this field, with collaboration primarily occurring within institutions and limited cross-institutional cooperation. Furthermore, these research groups have different areas of focus. Journals in this field can be classified into three categories: engineering safety journals, fire safety journals, and forest fire prevention and control journals. The research findings exhibit a multidisciplinary approach, and the articles generally maintain a high quality and carry influence in various disciplinary domains.

(3) The cluster analysis of subject terms reveals that the main research directions in the field of fire risk assessment at this stage are typical fire site risk, fire risk assessment methodology, forest fire and area fire assessment and prediction, and fire experiments and testing. The research hotspots primarily revolve around investigating fire and explosion accidents, assessing the vulnerability of fire subjects, and identifying potential fire hazards. The application of artificial intelligence technology is identified as a pivotal tool for future development. The analysis of highly cited outbreak literature highlights the significance of Westerling, A.L. (2006), Chuvieco, E. (2010), Martinez, J. (2009), Giglio, L. (2016), Giglio, L. (2018), and Guo, F.T. (2016) as important contributions to this field. It is recommended that scholars who are new to this field read these articles, which will greatly contribute to a rapid understanding of the field of fire risk assessment.

**Author Contributions:** Conceptualization, Z.T.; methodology, T.Z.; software, T.Z.; validation, L.W., T.Z. and S.R.; formal analysis, S.C.; investigation, Z.T.; resources, T.Z.; data curation, Z.T.; writing—original draft preparation, Z.T.; writing—review and editing, T.Z.; visualization, Z.T.; supervision, L.W.; project administration, T.Z.; funding acquisition, T.Z. All authors have read and agreed to the published version of the manuscript.

**Funding:** Financial support was given by the National Natural Science Foundation of Hebei Provence (No. E2023507001), the National Natural Science Foundation of China (No. 52174224, 51804314), the Key R&D Program of Hebei Province (No. 22375417D), and the Science and Technology Program of Fire and Rescue Department Ministry of Emergency Management (No. 2022XFCX19).

**Institutional Review Board Statement:** Not applicable.

**Informed Consent Statement:** Not applicable.

**Data Availability Statement:** No new data were created or analyzed in this study. Data sharing is not applicable to this article.

**Conflicts of Interest:** The authors declare no conflict of interest.

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
