# Peer review of "Knowledge Mapping for Fire Risk Assessment: A Scientometric Analysis Based on VOSviewer and CiteSpace"

_fire, doi:10.3390/fire7010023_

Round 1
Reviewer 1 Report
Comments and Suggestions for Authors
Dear authors,
The study submitted is a very interesting article that describes the continuous growth and evolving landscape of fire risk assessment research over the past five decades, revealing dominant forces such as China and the United States. The study highlights that the application of artificial intelligence technology is identified as a pivotal tool for future development.
In my opinion the paper should be published after major revisions.
The title is appropriate. It is of course essential to clearly explain the objectives of the work in the Introduction.
My suggestions are the following:
I suggest to the authors to expand the methodology section, providing more detailed explanations of the methods employed.
Line 63
Consider adding some references, such as Pacifico et al. 2023, who conducted a study on two fires—one clearly of anthropogenic origin and the other natural—highlighting the presence of associations of elements potentially toxic to human health in the ashes produced by the fires and deposited on the topsoil.
Line 130
Which statistical analyses? The authors need to be specified and explained.
Line 415-423
As there is a reference to a study here that analyses the propagation in smoke produced by fires for predictive purposes, I suggest incorporating references in the introduction section to studies that substantiate the transport and dispersion of ashes, even over considerable distances, such as Dimitrios E.A. 2020.
Reviewer 2 Report
Comments and Suggestions for Authors
This paper aims to explore the research hotspots and frontier trends in fire risk assessment and comprehend its macroscopic development trajectory. A sample of 1596 papers published from 1976 to 2023 were extracted from the Web of Science database. A knowledge map was created. Three methods including bibliometric methods, visual analysis, and content analysis were employed to uncover the research pulse and hotspots in the field, offering insights into its future development The results indicate that research in fire risk assessment has demonstrated continuous growth over the past 50 years. The dominant research forces and expert groups are listed. In addition, it is imperative to enhance the importance accorded to fire risk assessment, foster international and inter-institutional cooperation, and prioritize research innovation.
Comments on the Quality of English LanguageThe quality of English language need to be improved.
Round 2
Reviewer 1 Report
Comments and Suggestions for Authors
Dear authors,
thank you for incorporating all of my suggestions.
I believe the responses and additions are comprehensive and have enhanced the proposed work.